# Morphology Analysis of Friction Surfaces of Composites Based on PTFE and Layered Silicates

**DOI:** 10.3390/polym14214658

**Published:** 2022-11-01

**Authors:** Iuliia Kapitonova, Nadezhda Lazareva, Praskovia Tarasova, Aitalina Okhlopkova, Samuel Laukkanen, Vasiliy Mukhin

**Affiliations:** Federal State Autonomous Educational Institution of Higher Education “M. K. Ammosov North-Eastern Federal University”, 58 Belinskiy str, 677027 Yakutsk, Russia

**Keywords:** polytetrafluoroethylene, layered silicates, mechanical activation, friction coefficient, wear resistance, running-in, friction surface

## Abstract

In the present study, the tribological behavior of polytetrafluoroethylene (PTFE) composites filled with natural layered silicates (LS) was investigated. The change in the morphology of the friction surface of composites depending on the content and chemical composition of layered silicates has been shown. The friction surface of PTFE composites with layered silicates was investigated by scanning electron microscopy (SEM). The formation on the friction surface of a special layer with a structure different from the bulk of the polymer, which is formed from particles of fillers and wear products, has been established. The thickness of this layer is independent of the content of layered silicates in the polymer. It was indicated that wear debris of PTFE composites was assembled during friction and uniformly cover the friction surface layer by layer, thereby forming a protective layer.

## 1. Introduction

PTFE is a polymer that has various applications due to its unique combination of wide operating temperature range with low friction and low reactivity. However, despite the combination of unique properties, the main problem is the high wear rate of PTFE. This disadvantage greatly limits the use of PTFE and has become the subject of investigation of many researchers. The mechanism of wear reduction of PTFE-based composites using different fillers remains a hot topic of discussion. Effects of the wide range of different types of fillers on PTFE properties have been investigated for fibers [1,2], bronze [3], aluminum oxide [4], graphite [5], carbon nanotubes [6], and molybdenum disulfide [7].

The important advantage of the PTFE-metal sliding system is its ability to provide low friction and low wear under dry sliding conditions, which is called the self-lubricating effect. This happens due to the unique structure of PTFE macromolecules with a fluorine-coated carbon backbone. Many authors explain this effect mainly by the polymer composite’s ability to apply a transfer film to the opposite surface, as well as by the adhesive and chemical properties of this film [8,9,10]. However, this effect is possible not only in the formation of a protective structure but also under the condition of forming a shearing layer on the polymer surface that shifts relative to the counterbody and the rubbing part [11]. The formation of a shear layer is necessary to reduce the adhesion and various “plowing” interactions between surfaces moving relative to each other. Also, such layers reduce stresses in the bulk of the polymer. Unfortunately, there is not enough literature devoted to studying the processes on the polymer composite friction surface.

As a result of friction between two surfaces a complex process of multifunctional interaction occurs. This leads to a change in the structure and properties of the surface layers of materials. Such a change in structure and properties includes many interrelated processes that occur on the contact surfaces. According to the opinion of famous scientists, such as F. Bowden and D. Tabor, deformation of contact irregularities and interaction of materials at the molecular level can occur at the contacts of friction surfaces. Under appropriate conditions, depending on the load, friction conditions, properties of the contacting surfaces, and other factors, these processes can be accompanied by the dissipation of mechanical energy [12,13,14].

The polymer’s chemical nature has a significant effect on the processes of deformation and destruction of the friction surface. Therefore, these processes cause activation and change in the surface layers of the contacting parts. This determines the intensity of chemical transformations and physicochemical interactions of the abraded materials [15,16]. The intensity of tribochemical reactions seems to be the main factor in the system’s transition to a stationary state with a minimum level of wear [17,18,19].

All these phenomena occur in the process of adaptation of materials to friction conditions. These changes lead to the formation of a layer of the secondary structure as a result of friction of the polymer composite. This layer differs in physical, physicochemical, and thermophysical properties from the composite bulk properties. Most often, the process of material adaptation is considered a running-in process. The running-in process plays an important role in the study of the tribological properties of composite materials. The process determines the serviceability and wear resistance of technical polymeric materials and coatings [20,21]. It is necessary to choose combinations of components to further understanding of the effects of self-lubrication and self-organization of the structure of polymer composite materials during friction and wear. That will allow to ensure the lability of polymer structural elements to facilitate the structuring process during frictional loading.

Many aspects of the practical use of LS as PTFE fillers have not yet been fully studied, especially questions about the nature and physical nature of the interaction of polymer composite materials (PCMs)—with friction surfaces of parts. In addition to the previous works of the authors [17,19,22,23], in this article, an attempt was made to solve the issue of the mechanism of increasing wear resistance and the formation of a protective layer on the friction surface, depending on the friction conditions and the structure and composition of layered silicates. Using layered silicates as a filler leads to a significant improvement in properties at very low filler content, such as 5 wt.% [24]. Layered silicates are hydrophilic, which makes them very difficult to disperse uniformly in the polymer matrix. To ensure uniform distribution, the treatment of layered silicate with organic compounds is widely applied to replace the interlayer cations. The interlayer cations can be replaced, for example, by quaternary ammonium or phosphonium cations, preferably with long alkyl chains. Layered silicates with high cation exchange capacity are suitable for use in organic treatment [25,26,27], but the processing temperature of PTFE-based composites significantly exceeds the degradation temperature of all organic surfactants. There is a need to use other methods to increase the surface activity of layered silicates, e.g., mechanical activation and the use of functional additives. The choice of layered silicates as fillers for PTFE is explained, firstly, by their layered structure: they are able to form a shearing layer on the friction surface. Secondly, layered silicates are actively involved in tribochemical processes due to their chemical composition and provide a rapid transition to the stable flow of friction.

The aim of the work was to study the effect of layered mineral silicates on the tribological properties and the structure of the friction surface of PTFE composites.

This article presents the results of tribological testing of PTFE–layered silicate samples depending on the content and nature of fillers. The issue of the importance of the running-in stage of composites is highlighted. The SEM method shows the features of the formation of the morphology of the friction surface of PTFE/LS composites, taking into account the time of tribological tests, the type of filler, and its concentration.

## 2. Materials and Methods

### 2.1. Materials

PTFE of PN grade (GaloPolymer, Kirovo-Chepetsk, Russia) was used for the preparation of composite samples. A list of selected natural layered silicates used as fillers is presented in Table 1. Highly dispersed synthetic magnesium spinel was also used as a filler. The magnesium spinel functional additive is actively involved in tribochemical processes and ensures the formation of the surface layer during frictional contact. The prerequisite for the use of magnesium spinel was its structural activity, identified and investigated in previous work [22], as well as its chemical composition.

The structure of layered silicates is presented in Figure 1. The structure of layered silicates is based on tetrahedral silicon–oxygen and octahedral Al- (gibbsite) or Mg- (brucite) oxygen–hydroxyl networks. The tetrahedra in the silicon–oxygen network are linked by the vertices of their bases in a hexagonal pattern. In an octahedral grid, the octahedra are connected by their edges so that their centers also form a hexagonal pattern. Tetrahedral and octahedral networks of similar size articulate with each other into layers, which for each particular mineral represent a certain combination of these networks [28,29].

The nomenclature committee of AIPEA (International Association for the Study of Clays) subdivides layered silicates into two types: two-layer 1:1 (asymmetrical) and three-layer 2:1 (symmetrical). Asymmetrical types have serpentine and kaolinite, while symmetrical types have phlogopite, muscovite, vermiculite, and montmorillonite.

The magnesium spinel used in this work is a highly dispersed double oxide powder with the formula MgAl_2_O_4_. This MS was synthesized by the Institute of Solid State Chemistry and Mechanochemistry of the Siberian Branch of the Russian Academy of Sciences (Novosibirsk, Russia). The MS particle size is about 75 nm, specific surface area 170 m^2^/g, density 3600 kg/m^3^, melting temperature 2135 °C.

The structure of the MS is close to the closest cubic packing of oxygen the tetrahedral voids that are occupied by Mg^2+^ ions (radius 0.078 nm), the octahedral voids—A1^3+^ ions (radius 0.057 nm) [30]. The structure of MS is presented in Figure 2.

### 2.2. Sample Preparation

To remove adsorbed water, the initial powders of layered silicates, except for vermiculite, were dried in an oven ES-4610 (Saint Petersburg, Russia) for 6 h at a temperature of 105–120 °C. Vermiculite was treated in a different way: heat treatment at 950 °C for 3 s in an oven (Elsklo, Czech Republic). During the process, the release of water vapor and swelling of vermiculite occurs due to its splitting into separate mica plates slightly bonded to each other. The initial LS, activated LS, and activated MS were used for combination with composites. The mechanical activation of LS was used for grinding and increasing of its reactivity. Equipment for mechanical activation was an Activator 2S planetary mill (Activator, Novosibirsk, Russia), time 2 min, acceleration 80 g. Phlogopite was treated for 10 min to transform it to a highly dispersed state. Composite samples were obtained by mixing a polymer with a highly dispersed filler powder after mechanical activation, then cold molding and sintering. Mixing was carried out in a paddle mixer. The samples were sintered in the temperature range 370–380 °C. In the case of preparation samples with bentonite and vermiculite, the original (untreated) and mechanically activated LS were used.

### 2.3. Experimental Methods

The tribological characteristics of the composite samples were determined—the mass wear rate and the coefficient of friction. The tests were carried out on a UMT-3 universal tribometer (CETR, Mountain View, CA, USA) according to the following parameters: “pin-on-disk” friction scheme, load 2 MPa, sliding speed 0.25 m/s. The samples for the tribological study had the shape of a cylinder with a diameter of 10 mm and a height of 20 mm. The counterbody was a disk made of steel AISI 1045 with 56–58 HRS hardness of and Ra = 0.06–0.07 μm roughness.

Two types of tribological test were performed. In the first, samples were worn for 3 h (for PTFE containing muscovite, phlogopite, bentonite, vermiculite). The second test comprised two stages. Tribological tests were carried out, taking into account two modes: the running-in period and the normal wear period. For tests of the running-in period, the weight change was measured using the Mettler Toledo scale (Columbus, OH, USA) (±10 μg). The normal wear period test was performed within 5 h after the end of the previous test. The change in mass of the sample before and after testing, as well as the coefficient of friction, were recorded. The morphology of the friction surface of composites was studied using SEM (JSM-6480LV and JSM-7800F; JEOL, Tokyo, Japan).

Figure 3 shows the technological scheme for the development of samples and the methods of research applied.

## 3. Results and Discussion

### 3.1. Tribological Tests

Durability and reliability of engineering polymer materials and coatings largely depends on the running-in process in tribology. The running-in period is characterized by processes with minimal energy consumption, but involves changes in the microstructural state of the surface, wherein friction occurs along the contacting projections of the irregularities of both rubbing surfaces. A lot of irreversible processes occur on the surface of materials [31].

It was shown that introduction of layered silicates in PTFE significantly increased the wear resistance of PTFE (Table 2 and Table 3) up to 1000 times. The additional usage of MS as additives improved wear resistance of composites containing kaolinite and serpentine. Friction coefficients of these composites were reduced. It was found that the ratio of LS and MS depends on wear resistance. The addition of MS leads to a faster transition of the friction mode to the normal wear mode in the case of PTFE–serpentine composites. However, low MS content as additive provided the highest wear resistance in the steady-state friction mode. As a result, the optimal content of MS was revealed for PTFE–kaolinite composites that most effectively influenced tribological characteristics.

Table 4 shows the tribological characteristics of composites containing bentonite and vermiculite and compares them. A significant improvement in wear resistance was already achieved at 2 wt.%. In previous works [32,33], the filling up to 2 wt.% was defined as the “first critical concentration” when filling PTFE with nanosized ceramics. In this work, the statement is also valid in the case of filling with layered silicates.

However, it should be noted that in the case of vermiculite, the volume content corresponding to 2 wt.% of other layered silicates was already achieved with the introduction of ~1 wt.%, because of very low density (0.1–0.3 g/cm^3^). Therefore, for these composites, the maximum deformation-strength characteristics were achieved with the introduction of 1 wt.% vermiculite.

The effect of MS on composites with phlogopite (Figure 4) was generally not straightforward. The achievement of a significantly low rate of mass wear with the introduction of 0.5 wt.% of phlogopite without the addition of MS was noted. However, a significant decrease in wear resistance was observed with the additional introduction of 1 wt.% MS.

A possible explanation for this is the individual chemical composition of phlogopite that determines its physical characteristics, such as higher density, hardness, elasticity, and larger particle size, compared to other LS used in the work.

It is worth noting that the introduction of small amounts of MS reduced the wear resistance of composites compared with the wear resistance of composites containing only LS. This was especially noticeable in composites with muscovite and phlogopite with a concentration of LS up to 1 wt.% (Table 5). The results include a run-in period. Total friction time was 3 h.

In some composites with low LS content, it was noted that the introduction of a certain amount of magnesium spinel led to a decrease in the friction coefficient, but the level of wear resistance increased in comparison with other composites. Since the friction conditions of composites were the same (sliding speed, temperature, load), besides the composition of composites, a similar effect can be explained only by the properties of the forming surface layer differing from the layers that form on the surfaces of other composites. It is known [34] that an increase in surface hardness, ceteris paribus, helps to reduce the friction coefficient, which is associated with a decrease in real contact area (RCA) of rubbing surfaces. In this case, when the content of LS was not large, the presence of active MS contributed to the formation of a denser and harder surface layer with a smaller RCA, but with a high adhesive bond with the subsurface layer. This circumstance, however, reduced the wear resistance of the material, because it prevented the ease of sliding of the sublayers relative to each other. From Table 5 it can be seen that in composites with muscovite, the introduction of 0.1 wt.% MS was enough to form a surface layer of that kind, and in the case of the introduction of phlogopite, a similar layer was formed when the ratio between phlogopite and MS was 1:1. Thus, there is an assumption about the influence of the chemical composition of phlogopite and muscovite on the appearance of this effect. Phlogopite and muscovite differ in the composition of the cations of octahedral grids (see Table 1), i.e., the conditions for the formation of the surface layer with low friction coefficient, but with relatively high wear is: PTFE–muscovite (octahedral grid includes Al^3+^) + 0.1 wt.% MS and PTFE–phlogopite (octahedral grid includes Mg^2+^) + MS in the ratio of 1:1.

It should be noted that during the mechanical activation of layered silicates, their octahedral grids were destroyed first of all. There are many works that provide evidence that as a result of the mechanical activation of LS, the cations of octahedral grids are released and become active centers representing coordination unsaturated ions [35,36,37,38,39]. Considering the known aluminum atom activity towards the fluorine atom, it becomes obvious that the cause of the formation of an undesirable increase in material wear is the aluminum cation of the muscovite octahedral grids.

An increase in the LS content eliminates this effect because the larger number of silicate particles reduces the concentration of aluminum cations. Therefore, it increases the lability and mobility of the surface protective layer, which leads to the significant reduction in the wear of composites, but slightly increases the friction coefficient. The reason for the decrease in the friction coefficient of composites containing MS compared with composites without MS is the activity of nanoparticles and their ability to cluster. This generally strengthens the formed layer on the friction surface.

### 3.2. Morphology Analysis

To confirm these assumptions, we studied the structure of the friction surface of composites. As shown by SEM, the running-in period is characterized by changing of microroughness of the friction surface and forming of optimal surface roughness. In this case, the surface layers undergo plastic deformation, and the structure of the material becomes looser and more amorphous. An increase in temperature is also observed. The above processes lead to the formation of a layer on the friction surface, which differs in physicochemical, mechanical, and thermophysical parameters from the bulk of the polymer [10]. The formed layer can be called a layer of secondary structure. Under the influence of elevated temperatures, a mobile dissipative surface structure of the composite is formed. Therefore, according to the authors [40,41,42], this explains the increase in the wear resistance of composites.

A secondary structure layer was formed on the friction surface by composites containing layered silicates, as shown by SEM. This layer is visually different from the volume of the composite, as shown in Figure 5. The layer was a fine mass and can be formed from wear particles of the material. Perhaps this layer acts as a solid lubricant that protects the material from abrasion and wear.

In [43], the authors studied PTFE–aluminum oxide composites and considered mechanisms for improving wear resistance. According to the authors, during the friction of the composite, a transfer film is formed on the counterbody and multiple wear particles are formed. The circulation of such wear particles between the tribopairs is the main factor maintaining the ultralow wear of composites. The formation of wear particles is accompanied by significant changes in the structure and composition of the surface layer. The presented wear mechanism for composites includes such processes as dispersion, defragmentation, and oxidation [44,45,46]. Layered silicate filler particles accumulate on the friction surface and orientate along the direction of friction. It is known that graphite can orientate under friction under load. Similarly, the particles of layered silicates can also be oriented so that the principal axes become parallel to the direction of sliding [47]. The formed layer has low shear stability and can slip relative to the counterbody surface and the composite surface in the shear direction. In addition, the protective layer is plastic, and it smoothens the friction surface, as shown in Figure 6.

In the case of the formation of a continuous protective layer due to its softer consistency, cracks and layers were formed. They formed on each other perpendicular to the direction of friction, localizing the shear deformations of the subsurface layer.

Comparison of the friction surfaces of composites containing bentonite as a filler (Figure 7 and Figure 9) revealed a significant difference depending on mechanical activation. The microprotrusions were visible in composites with unactivated bentonite. It is likely that bentonite agglomerates do not wear out during friction and protrude above the surface. The direction of shear deformations is clearly visible in the SEM images due to the fragility of the layer. The edges of the torn layer were slightly carried in the direction of shear to the surface of the front layer. Thus, the surface of the composites resembles scales.

A similar type of wear occurs during the oxidative wear of metals, when plastic oxide films of a secondary structure (structures of the first kind) are formed on the metal surface as a result of structural adaptation to abrasion. Such structures wear out by moving thin films along the contact surface with their subsequent removal from the surface (Figure 8) [47].

The surface of composites with mechanically activated bentonite (Figure 9) was smoother without microcracks. It is possible that mechanically activated particles of silicates contribute to the formation of a stronger protective layer.

The surface of composites after friction containing vermiculite showed the same changes as bentonite (Figure 10 and Figure 11), depending on mechanical activation, but unlike composites containing bentonite, the consistency of the layer was obviously more fragile and less ductile. The surface of the layer was characterized by the formation of many cracks. In addition, due to the nonplasticity of this layer, no layering was observed on the surface of the composites with bentonite. The mechanical activation of vermiculite did not lead to any visible changes in the structure or nature of the formation of protective layers. It seems that the consistency of these layers depends on the chemical composition of the layered silicates.

The protective layer formed on the surface of composites containing vermiculite is suitable for describing brittle secondary structures, which also arise during oxidative wear (a structure of the second kind), given by Kostetskiy in [47]: “Fragile films of secondary structures…under the influence of normal and tangential stresses are first covered by a network of cracks and then brittlely peeled off from the friction surfaces.” The scheme of formation and destruction of secondary brittle structures is shown in Figure 12.

The author claims that “as the particles of the protective layer are removed, the surface of the composite under the protective layer comes into contact with the counterbody, which leads to the formation of a new protective layer” [47].

The formation of the protective layer observed after abrasion of the composite with serpentine for 1 h and 5 h after the running-in mode showed a big difference (Figure 13). After 1 h of abrasion, the friction surface was covered with cracks. A magnification of 10,000 allows us to see that fragments of the surface layer are interconnected by PTFE fibrils.

A similar SEM image was presented in [47]: the authors used scanning electron microscopy to study the friction surfaces of composites with an ultralow wear rate of 10^−7^ mm^3^/(N∙m) based on PTFE and 5 wt.% Al_2_O_3_ with a dispersion of 80 nm. The authors described SEM image data: “The friction surface of the composite is characterized by the presence of a network of cracks. The areas separated by cracks are about 10 μm in size and, apparently, can be easily removed from the surface. However, neither voids nor particles of wear products were found on the studied surfaces. At a higher magnification it can be seen that the cracks are filled with PTFE fibrils, which most likely prevent the removal of fragments from the friction surface in the form of wear particles” [48].

Another surface image (Figure 14) of the composite with vermiculite after abrasion for 1 h allowed us to see the formation of the secondary structure layer by wear particles in the form of cluster formations.

The average size of each cluster was 250 nm. It is clearly visible that the clusters cover the surface in layers and the clusters are located in each layer, maintaining a certain order. In the second image, the area of the lower layer is highlighted on the right.

Figure 15 shows the topography of the friction surface of PCM containing only LS and LS with MS for comparison. It can be seen from the images that MS is involved in the friction process and affects the morphology of the friction surface. As we can see, the friction surface of the PTFE/LS+MS composites has a dense and even structure. The visual difference is confirmed by the difference in the coefficients of friction of the composites. Comparison of the morphology of the friction surface of composites containing kaolinite and serpentine does not show significant differences; however, in PTFE–serpentine + MS, the friction surface is visually smoother.

The following SEM images (Figure 16) allow us to judge the thickness of the protective layer formed on the friction surface with the introduction of MS. The first SEM image shows that when the sample was cleaved, fragments of the protective layer hung on the PTFE fibrils. Unfortunately, it was not possible to prove the nature of these fragments using elemental analysis. It was not possible to detect these fragments after sputtering the sample surface with a conductive gold coating necessary for elemental analysis. Nevertheless, the smooth edges of the fragments that were absolutely not characteristic of the configuration of the edges of mechanically activated LS (Figure 16c) and dispersed MS powders allow us to assume that these are fragments of the protective surface layer formed in the process of abrasion of the composite.

This circumstance confirms the assumption of the authors [47] that the surface layer can easily separate from the surface of the material under it, and the fragments are interconnected by PTFE fibrils. The edges of the fragments are even, which confirms the density of this layer. In the second image, the boundary between the main material and the surface layer is clearly visible. The layer thickness in both images is approximately the same and equal to ~1 μm, although the content of fillers in the composites is noticeably different. Therefore, it can be assumed that after a certain abrasion time (5 h), the thickness of the layer of the secondary structure necessary to establish a stationary regime becomes constant, regardless of the filler content.

## 4. Conclusions

It was shown that the circulation of material wear particles and mechanochemical processes on the friction surface form the protective layer. The evolution of the secondary structure formation as the protective layer includes several stages. The first stage is associated with the deformation and loosening of the friction surface. The second stage includes wear of the loosened part of the composite surface and its dispersion. The third stage consists of the formation of clusters uniformly covering the friction surface layer by layer. The fourth stage is associated with the adjustment and recovery of the layer in the case of layer fragments, as well as with the layer damage because of ingress of solid impurities.

The formed layer has sufficiently high strength and wear resistance. This protective layer has a complex chemical composition. However, the improvement of the above properties is influenced by the metal cations of the fillers, according to the authors. The composition and structure of layered silicates significantly influence the formation of this wear layer. The dispersibility of LS is one of the important properties for this.

From all this, it follows that due to their structure and composition, layered silicates are convenient components of a composite material for implementation of the potential ability of a material to self-adapt under frictional interaction.

## Figures and Tables

**Figure 1 polymers-14-04658-f001:**
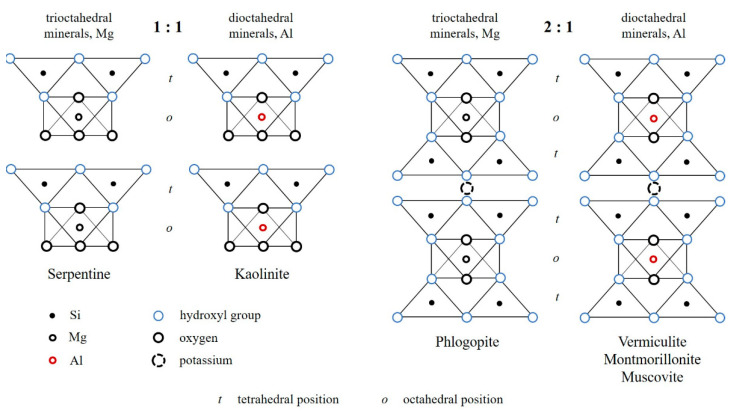
Structure of layered silicates.

**Figure 2 polymers-14-04658-f002:**
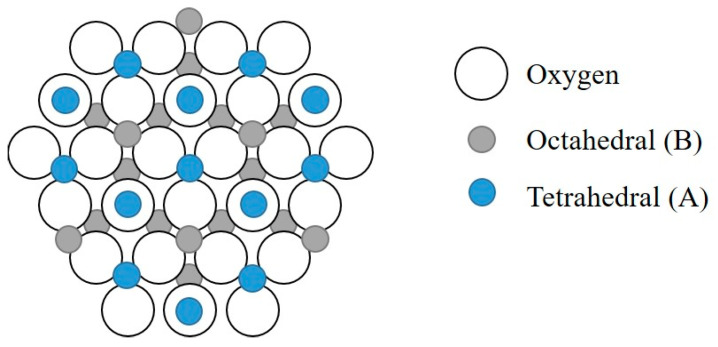
Magnesium spinel structure.

**Figure 3 polymers-14-04658-f003:**
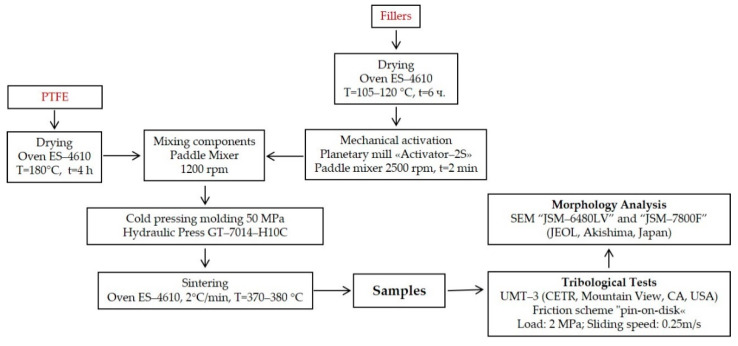
Technological scheme for the development and testing of polymer composites.

**Figure 4 polymers-14-04658-f004:**
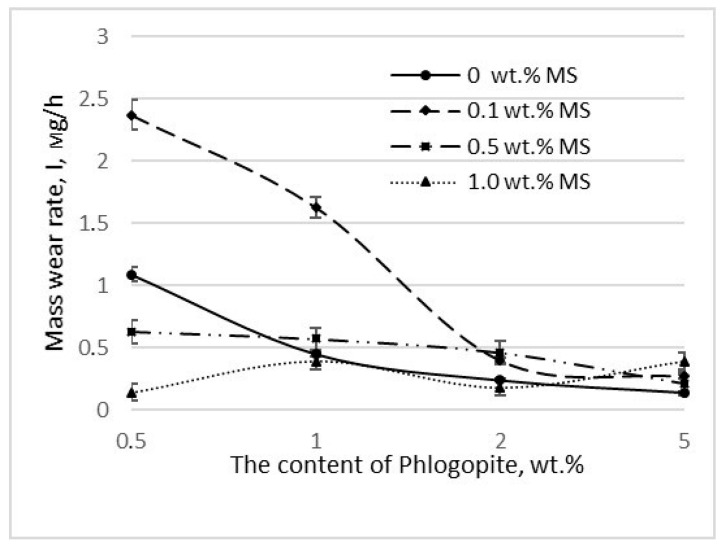
The wear resistance of composites containing phlogopite, depending on the concentration of MS.

**Figure 5 polymers-14-04658-f005:**
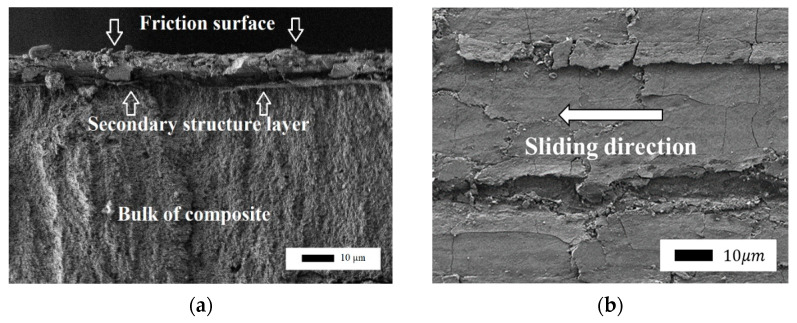
The structure of the friction layer of the PTFE–kaolinite composite: (**a**) cut of the friction surface in the lateral projection (1000× magnification); (**b**) top view, the arrow shows the sliding direction of the counterbody (1000× magnification).

**Figure 6 polymers-14-04658-f006:**
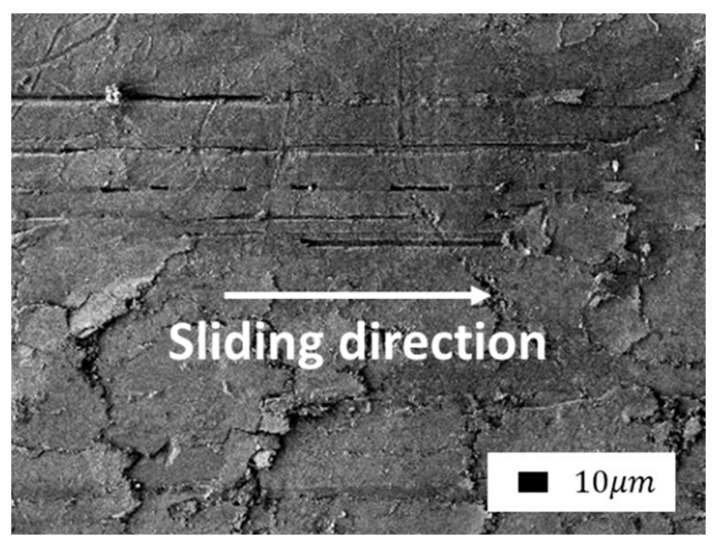
SEM image of the friction surface of a composite filled with serpentine (500× magnification).

**Figure 7 polymers-14-04658-f007:**
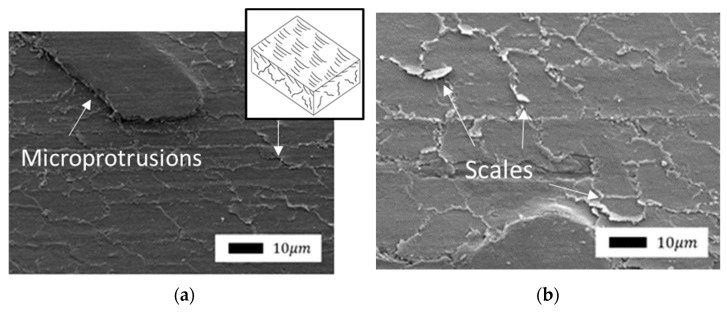
SEM images of the friction surface of composites containing unactivated bentonite: (**a**) 1.0 wt.%; (**b**) 2.0 wt.% (1000× magnification).

**Figure 8 polymers-14-04658-f008:**
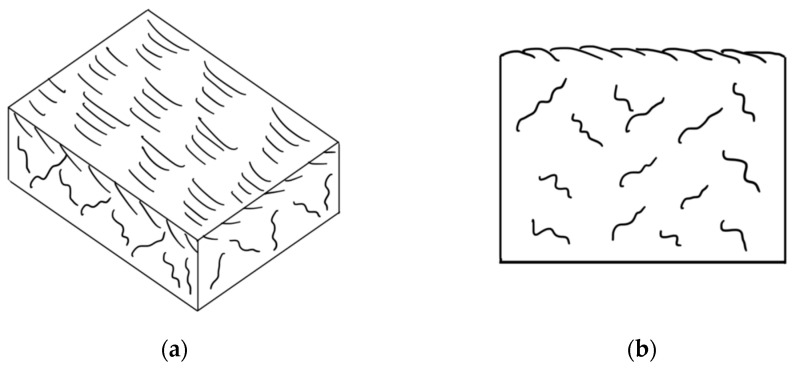
Scheme of the formation and removal from the surface of plastic secondary structures of the first kind: (**a**) top view; (**b**) section.

**Figure 9 polymers-14-04658-f009:**
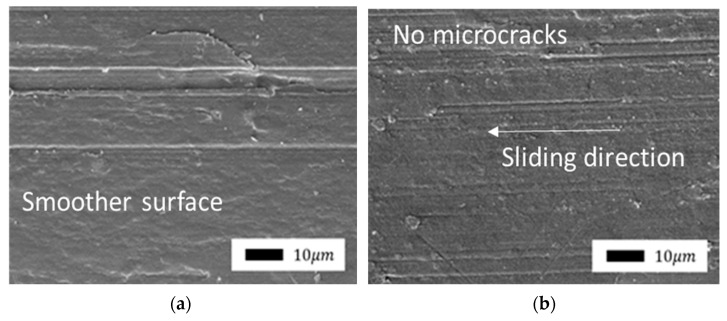
SEM images of the friction surface of composites containing mechanically activated bentonite: (**a**) 1.0 wt.%; (**b**) 2.0 wt.% (1000× magnification).

**Figure 10 polymers-14-04658-f010:**
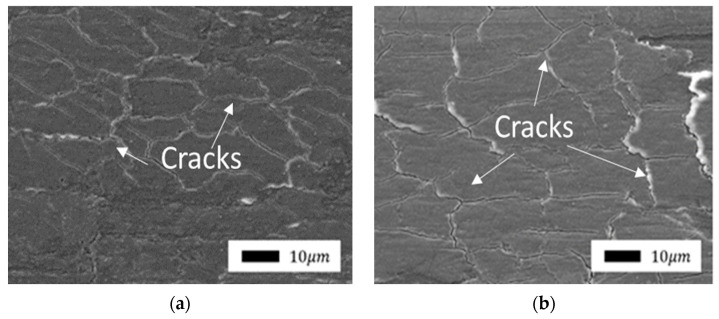
SEM images of the friction surface of composites containing nonactivated vermiculite: (**a**) 1.0 wt.%; (**b**) 2.0 wt.% (1000× magnification).

**Figure 11 polymers-14-04658-f011:**
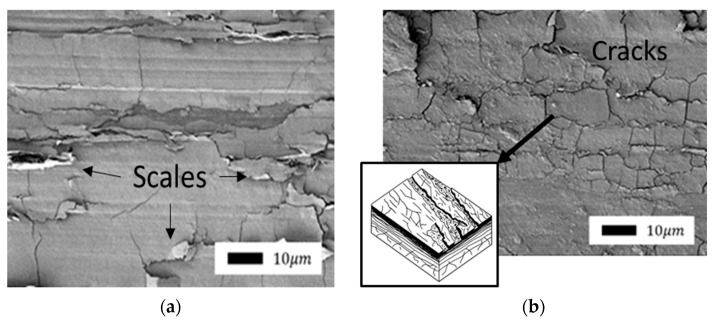
SEM images of the supramolecular structure of composites containing mechanically activated vermiculite: (**a**) 1.0 wt.%; (**b**) 2.0 wt.% (1000× magnification).

**Figure 12 polymers-14-04658-f012:**
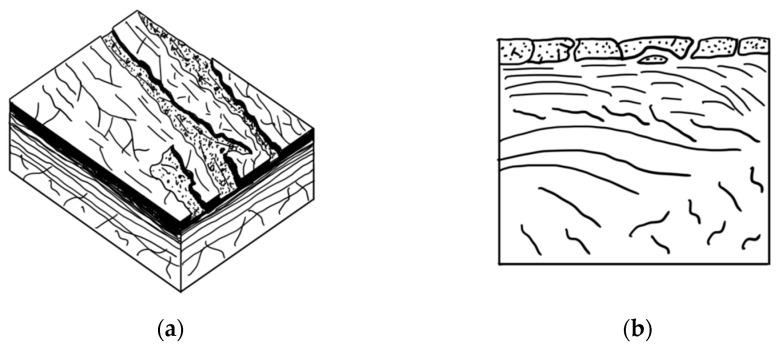
Scheme of the formation and destruction of secondary brittle structures: (**a**) top view; (**b**) section.

**Figure 13 polymers-14-04658-f013:**
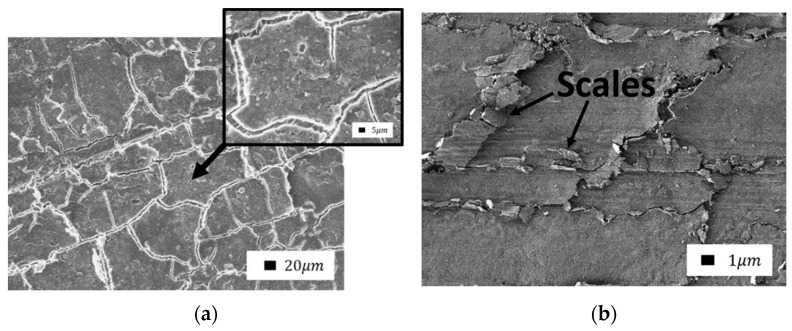
SEM images of the friction surface of a composite containing 2.0 wt.% serpentine: (**a**) after 1 h of friction in a quasi-stationary mode (2500× magnification); (**b**) after 5 h of friction (3000× magnification).

**Figure 14 polymers-14-04658-f014:**
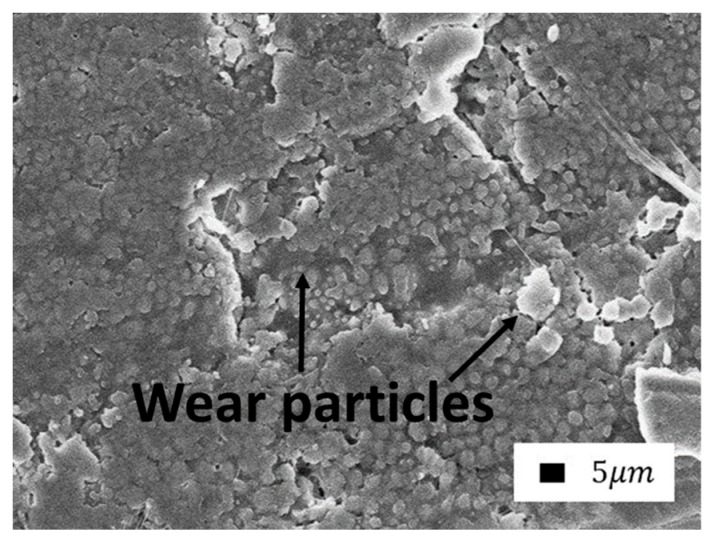
SEM images of the secondary structure layer on the friction surface (10,000× magnification).

**Figure 15 polymers-14-04658-f015:**
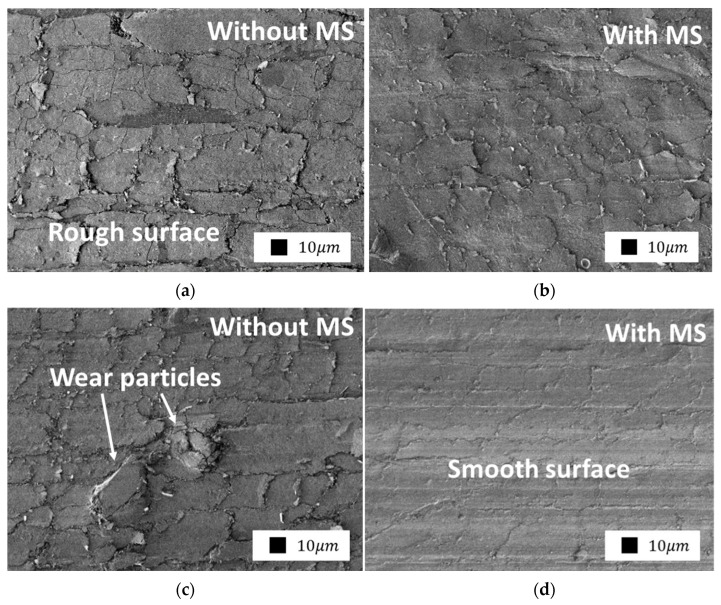
SEM images of the composite’s friction surface containing: (**a**) 2.0 wt.% kaolinite; (**b**) 1.8 wt.% kaolinite and 0.2 wt.% MS; (**c**) 2.0 wt.% serpentine; (**d**) 1.8 wt.% serpentine and 0.2 wt.% MS (500× magnification).

**Figure 16 polymers-14-04658-f016:**
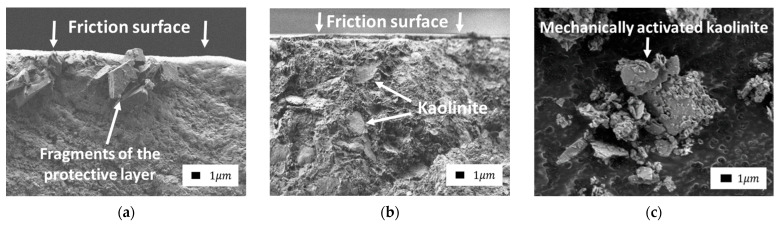
SEM images of the transverse cleavage of composites with kaolinite: (**a**) PTFE + 0.9 wt.% kaolinite + 0.1 wt.% MS; (**b**) PTFE + 4.0 wt.% kaolinite + 1.0 wt.% MS (3000× magnification); (**c**) SEM image of mechanically activated kaolinite (5000× magnification).

**Table 1 polymers-14-04658-t001:** Classification and deposits of the layered silicates used.

Mineral	Classification	Deposit	Structural Formula	Laying Type	Cations in Tetrahedra	Octahedron Cations	Interlayer Cations
Kaolinite	polymineral, aluminum hydrosilicate	Glukhovetskoye Ukraine	Al_2_[Si_2_O_5_](OH)_4_	1:1	Si^4+^	Al^3+^	-
Serpentine	mineral, magnesium hydrosilicate	Rikolatva Murmansk region, Russia	Mg_3_[Si_2_O_5_](OH)_4_	1:1	Si^4+^	Mg^2+^	-
Vermiculite	mineral, aluminum, magnesium and iron hydroaluminosilicate (hydromica)	Inaglinskoye Yakutia, Russia	K_2_(Al,Mg,Fe)_6_[A1_2_Si_6_O_20_](OH)_4_	2:1	Si^4+^, Al^3+^	Al^3+^, Mg^2+^, Fe^2+^	M^2+, 1^·nH_2_O
Phlogopite	mineral, magnesium aluminum silicate (mica)	Emelgakskoye Yakutia, Russia	KMg_3_ [AlSi_3_O_10_](F,OH)_2_	2:1	Si^4+^, Al^3+^	Mg^2+^	K^+^
Muscovite	mineral, aluminum aluminosilicate (mica)	Rikolatva, Murmansk region	KAl_2_ [AlSi_3_O_10_](OH)_2_	2:1	Si^4+^, Al^3+^	Al^3+^	K^+^
Bentonite (montmorillonite)	polymineral hydroaluminosilicate contains ~ 70% montmorillonite	Dashukovskoye Ukraine	(Na,Ca)_<0,4_(Al,Mg,Fe)_2–3_ [(Si,Al)_4_O_10_](OH)_2_*nH_2_O	2:1	Si^4+^, Al^3^	Al^3+^, Fe^3+^, Mg^2+^, Fe^2+^	Complex^n+, 2^

^1^ M^2+^: Mg^2+^, Ca^2+^; ^2^ Complex^n+^: Na^+^, Li^+^, K^+^, Rb^+^, Ca^2+^, Mg^2+^, Co^2+.^

**Table 2 polymers-14-04658-t002:** Tribological characteristics of composites based on PTFE and serpentine depending on the content of magnesium spinel.

Serpentine, wt.%	MS, wt.%	I_run-in_ ^1^, mg/h	I_st._ ^2^, mg/h	f ^3^
0	0	112.5 ± 3.4	65.5 ± 2.0	0.23 ± 0.007
2.0	0	1.8 ± 0.05	0.1 ± 0.003	0.32 ± 0.009
1.5	0.5	0.8 ± 0.02	0.2 ± 0.005	0.25 ± 0.007
1.8	0.2	0.9 ± 0.02	0.1 ± 0.003	0.24 ± 0.007
5.0	0	0.6 ± 0.01	0.1 ± 0.003	0.32 ± 0.009
4.0	1.0	0.4 ± 0.01	0.1 ± 0.003	0.26 ± 0.008
4.5	0.5	0.9 ± 0.02	0.1 ± 0.003	0.26 ± 0.008
4.8	0.2	1.0 ± 0.03	0.1 ± 0.003	0.27 ± 0.008

^1^ I_run-in_—the rate of mass wear during the run-in period; ^2^ I_st._—the rate of mass wear during the period of normal wear; ^3^ f—the friction coefficient.

**Table 3 polymers-14-04658-t003:** Tribological characteristics of composites based on PTFE and kaolinite depending on the content of magnesium spinel.

Kaolinite, wt.%	MS, wt.%	I_run-in_ ^1^, mg/h	I_st._ ^2^, mg/h	f ^3^
0	0	112.5 ± 3.4	65.5 ± 2.0	0.23 ± 0.007
2.0	0	1.9 ± 0.06	0.2 ± 0.005	0.33 ± 0.009
1.5	0.5	1.0 ± 0.03	0.1 ± 0.003	0.24 ± 0.007
1.8	0.2	1.1 ± 0.03	0.2 ± 0.005	0.22 ± 0.006
5.0	0	0.8 ± 0.02	0.1 ± 0.003	0.32 ± 0.009
4.0	1.0	1.6 ± 0.04	0.2 ± 0.005	0.27 ± 0.008
4.5	0.5	0.1 ± 0.01	0.1 ± 0.003	0.27 ± 0.008
4.8	0.2	0.5 ± 0.01	0.1 ± 0.003	0.23 ± 0.007

^1^ I_run-in_—the rate of mass wear during the run-in period; ^2^ I_st._—the rate of mass wear during the period of normal wear; ^3^ f—the friction coefficient.

**Table 4 polymers-14-04658-t004:** Tribological characteristics of composites containing vermiculite and bentonite.

Sample	Layered Silicates Content, wt.%	I_run-in_ ^1^, mg/h	I_st._ ^2^, mg/h
PTFE	0	112.5 ± 3.4	65.5 ± 2.0
PTFE + Bentonite	1	2.39 ± 0.07	2.50 ± 0.07
	2	1.03 ± 0.03	0.59 ± 0.02
	5	0.26 ± 0.01	0.17 ± 0.01
	7	0.17 ± 0.01	0.15 ± 0.01
PTFE + Vermiculite	1	2.15 ± 0.06	0.41 ± 0.02
	2	1.22 ± 0.04	0.37 ± 0.01
	5	0.66 ± 0.02	0.48 ± 0.01
	7	0.70 ± 0.02	0.51 ± 0.01

^1^ I_run-in_—the rate of mass wear during the run-in period; ^2^ I_st._—the rate of mass wear during the steady state period.

**Table 5 polymers-14-04658-t005:** Tribological characteristics of composites containing LS depending on their relationship with the MS.

Content of Fillers, wt.%		I ^1^, mg/h	f ^2^	Content of Fillers, wt.%		I ^1^, mg/h	f ^2^
Muscovite	MS			Phlogopite	MS		
0.5	0	5.15 ± 0.15	0.22 ± 0.006	0.5	0	4.18 ± 0.12	0.24 ± 0.007
	0.1	10.22 ± 0.30	0.19 ± 0.005		0.1	1.76 ± 0.05	0.26 ± 0.008
	0.5	2.18 ± 0.06	0.24 ± 0.007		0.5	9.50 ± 0.28	0.18 ± 0.005
	1.0	2.90 ± 0.09	0.25 ± 0.007		1.0	1.69 ± 0.05	0.22 ± 0.006
1.0	0	3.43 ± 0.10	0.24 ± 0.007	1.0	0	1.20 ± 0.03	0.25 ± 0.007
	0.1	9.88 ± 2.96	0.19 ± 0.005		0.1	1.84 ± 0.05	0.23 ± 0.007
	0.5	4.97 ± 0.14	0.21 ± 0.006		0.5	2.57 ± 0.08	0.22 ± 0.006
	1.0	4.70 ± 0.14	0.20 ± 0.006		1.0	3.79 ± 0.11	0.18 ± 0.005

^1^ I—rate of mass wear; ^2^ f—friction coefficient.

## Data Availability

Not available.

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
