# Peer review of "Morphology Analysis of Friction Surfaces of Composites Based on PTFE and Layered Silicates"

_polymers, 2022, doi:10.3390/polym14214658_

Round 1

Reviewer 1 Report

the work is well organized with enough experimental content. some suggestion might be useful.

1. the title is too general.  it should be more concrete, such as the surface morphology analysis?

2. in section 1 of the last paragraph, the content and method of the article or the organization constructure should be introduced.

3. no table 1. 

4. it is suggested to provode a detailed schematic diagram for the experiments.

5. more illustrations should be given in the SEM figures from Fig.5 To the end.

6. languages should be improved.

7. the conclusions should be reorganized with more clear structure. the first paragraph might fail to produce positive feedback to the readers.

Author Response

Thank you for your comments. We have attached our answers in a file below. Please see the attachment.

Reviewer 2 Report

This study investigates the tribological behavior of PTFE composites filled with natural layered silicates. The study has shown the influence of the concentration and chemical nature of layered silicates on the tribological characteristics and structure of the friction surface.

The study is interesting because it found out that layered silicates are convenient components of composite material for the implementation of the potential ability of a material to self-adapt under frictional interaction due to their structure and composition characteristics.

Therefore, the manuscript can be accepted for publication. However, a few minor revisions are required:

1. All acronyms need to be explained at their first appearance. Please check the entire manuscript.

2. In the introduction section, please highlight the novelty of the research compared to the previous studies. Additionally, the main conclusions of the research should be briefly highlighted.

3. The citation of Table 1 needs to be double-checked.

4. In subsection 2.3, it is better to present a schematic diagram for the experiment to make it easier for the reader to visualize.

5. The content presented in section 3 is too long. It is better to break down section 3 into a number of small subsections.

6. Finally, there are many English spelling and grammar errors throughout the manuscript. Therefore, correcting English spelling and grammar is mandatory. 

Author Response

(The authors gave the same response as above.)
